# Brain Natriuretic Peptide Biomarkers in Current Clinical and Therapeutic Scenarios of Heart Failure

**DOI:** 10.3390/jcm11113192

**Published:** 2022-06-02

**Authors:** Gianmarco Alcidi, Giovanni Goffredo, Michele Correale, Natale Daniele Brunetti, Massimo Iacoviello

**Affiliations:** 1Department of Medical and Surgical Sciences, University of Foggia, Viale Luigi Pinto 1, 71122 Foggia, Italy; gianmarco.alcidi@gmail.com (G.A.); michele.correale@libero.it (M.C.); natale.brunetti@unifg.it (N.D.B.); 2Cardiology Unit, Polyclinic University Hospital Riuniti of Foggia, Viale Luigi Pinto 1, 71122 Foggia, Italy

**Keywords:** brain natriuretic peptide, heart failure, diagnosis, prognosis, therapy

## Abstract

Brain natriuretic peptide (BNP) and its inactive N-terminal fragment, NT-proBNP, are serum biomarkers with key roles in the management of heart failure (HF). An increase in the serum levels of these peptides is closely associated with the pathophysiological mechanisms underlying HF such as the presence of structural and functional cardiac abnormalities, myocardial stretch associated with a high filling pressure and neuro-hormonal activation. As BNP and NT-proBNP measurements are possible, several studies have investigated their clinical utility in the diagnosis, prognostic stratification, monitoring and guiding therapy of patients with HF. BNP and NT-proBNP have also been used as criteria for enrollment in randomized trials evaluating the efficacy of new therapeutic strategies for HF. Nevertheless, the use of natriuretic peptides is still limited in clinical practice due to the controversial aspect of their use in different clinical settings. The purpose of this review is to discuss the main issues associated with using BNP and NT-proBNP serum levels in the management of patients with HF under current clinical and therapeutic scenarios.

## 1. Introduction

Brain natriuretic peptide (BNP) is a 32 amino acid peptide named after its initial detection in the porcine brain [1]. The highest concentrations are found in the ventricles, where BNP is produced from the proteolysis of a 108 amino acid precursor, proBNP. The secreted active BNP hormone and the inactive N-terminal fragment, NT-proBNP, can both be measured in serum [2,3]. The secretion of BNP is influenced by several pathophysiological conditions that are mainly associated with the presence of a cardiac dysfunction, myocardial stretch and high filling pressure as well as neuro-hormonal activation [3,4]. The biological activity of BNP tends to counterbalance the pathophysiological mechanisms leading to the onset and progression of heart failure (HF) by promoting natriuresis and diuresis and by inhibiting neuro-hormonal activation and cardiac remodeling [5].

Therefore, the serum measurement of these biomarkers has been investigated to identify patients with HF or a high HF risk as well as to stratify the prognosis and monitor therapy [3]. The possibility of using BNP as an HF therapy has been evaluated, either by administering a recombinant BNP peptide [6] or by inhibiting its degradation [7,8]. In recent years, BNP and NT-proBNP serum measurements have been used as criteria for patient enrollment in trials evaluating the efficacy of therapeutic strategies for HF [7,8,9,10,11,12,13,14,15].

Nevertheless, in routine clinical practice, the use of NPs is still limited by a few controversial aspects. The purpose of this review is to focus on the main issues associated with the dosage of BNP and NT-proBNP serum levels in the management of patients with HF in order to improve their use in routine clinical practice.

## 2. Variability of BNP and NT-proBNP Measurements

The dosage of NT-proBNP and BNP relies on the use of measurement kits available from various companies [3,16]. The dosage can also be based on traditional devices or point-of-care testing. The intrinsic variability in NT-proBNP and BNP values, the differences across the measurement techniques and the presence of biological factors that alter their plasma serum levels should be considered for the correct interpretation of the detected levels of natriuretic peptides (NPs) [3].

### 2.1. Measurement Variability

The secretion of NPs is associated with various factors, both hemodynamic and neuro-hormonal, that result in an intrinsic variation in NP plasma levels. The coefficients of variation for several measurement kits are approximately 30% [3]. This finding is particularly relevant, given that variations in peptide levels might be used to evaluate the therapy response and/or identify hemodynamic instability, as reported below.

### 2.2. NT-proBNP and BNP: Serum Levels and Clinical Correlates

Another essential aspect in interpreting the serum levels of NT-proBNP and BNP is the differences between their basal and pathological values as well as in the factors that influence their plasma levels [3,4,17]. First, the scale of the serum levels of BNP and NT-proBNP differs because of their contrasting half-lives (120 min for NT-proBNP vs. 20 min for BNP) and clearance rate [3]. Therefore, the mean serum levels of NT-proBNP are higher than those of BNP. Among the other factors influencing NP serum levels, body mass index and renal function play key roles [3,18].

NP serum levels are also influenced by age; the greater the age, the higher the serum levels [18]. Serum levels are also influenced by an altered cleavage [18]. When BNP is considered, a greater body mass index is particularly associated with lower serum levels. This is due to the fact that the clearance of BNP is associated with the activity of endopeptidase and the type C NP receptor—which are mainly present in adipose tissue, particularly in the obese population [4]. For this reason, these patients may have lower plasma BNP values than the normal population. The clearance of BNP and NT-proBNP is mediated by renal excretion. In patients with chronic kidney disease, a decreased estimated GFR is associated with elevated plasma BNP and an even greater elevation in NT-proBNP concentrations, whose clearance is more dependent on renal excretion.

Finally, beyond the parameters associated with cardiac function and filling pressure [19], atrial fibrillation (AF) is associated with elevated NPs.

These aspects should be considered together to optimize the use of the serum levels of NPs in routine clinical practice.

## 3. Natriuretic Peptides Defining Heart Failure Risk

The relevance of BNP and NT-proBNP has been evaluated both to assess HF risk and to determine the diagnosis. The recent universal definition of HF classifies the disease stages, including those preceding an HF diagnosis; i.e., stage A in patients with risk factors for HF and stage B in patients with asymptomatic cardiovascular diseases such as left ventricular hypertrophy, a chamber enlargement, a myocardial tissue abnormality or valvular heart disease [20].

In these stages, elevated NT-proBNP and BNP NP serum levels may indicate an elevated risk of HF occurrence [21,22]. This finding may aid in HF prevention and is particularly relevant in type 2 diabetes (T2DM). In patients with T2DM without baseline HF, a NT-proBNP level >125 pg/mL is associated with a higher risk of incident heart failure [22]. This finding is highly relevant, given that new emerging pharmacological approaches have been beneficial in decreasing HF risk in patients with T2DM, with or without a known cardiovascular disease [23,24,25].

A substudy of the CANVAS trial [26] evaluated the associations of baseline NT-proBNP with cardiovascular, renal and mortality outcomes as well as the changes associated with the administration of canaglifozin, a type 2 sodium–glucose cotransporter inhibitor (SGLT2i). NT-proBNP was assessed in a subgroup of enrolled patients at a baseline and at 1 and 6 years. A baseline NT-proBNP level > 125 pg/mL was significantly associated with incident HF hospitalization (HHF) (hazard ratio (HR): 5.40; 95% CI: 2.67–10.9) as well as with HHF/cardiovascular death (HR: 3.52; 95% CI: 2.38–5.20) and death due to all causes (HR: 2.53; 95% CI: 1.78–3.61). In patients treated with canaglifozin, HHF was lower than that in the placebo group. Accordingly, patients treated with canaglifozin showed an 11% decrease in NT-proBNP whereas an increase was observed in patients treated with the placebo. However, the proportion of patients with high NT-proBNP was greater than that of patients experiencing HHF during follow-up; the canaglifozin-mediated decrease in NT-proBNP explained only a small portion of the effects of canaglifozin in decreasing HF events.

## 4. Natriuretic Peptides in the Definition and Diagnosis of Heart Failure

The recent universal definition of HF includes NPs among the useful diagnostic criteria. In the definition of the syndrome, beyond the typical signs and symptoms caused by structural and/or functional cardiac abnormalities, elevated BNP or NT-proBNP levels can support the diagnosis [18]. Recent ESC guidelines [27] also recommended using NPs for the diagnosis of HF. In this setting, two relevant points should be considered: the recommended cut-offs and the differences between chronic and acute settings.

ROC curves for the diagnosis of HF from both BNP and NT-proBNP show a higher sensitivity and negative predictive value at lower peptide levels whereas higher peptide levels are associated with a higher specificity [3,28,29]. The ESC guidelines recommend the use of lower cut-offs to maximize the negative predictive accuracy. Consequently, in diagnosis of HF, the presence of BNP and NT-proBNP above the recommended upper limit should be confirmed by evidence of functional and structural echocardiographic cardiac abnormalities. This practice is relevant, given that the above factors influencing NP plasma levels such as age and AF can also be confounding factors affecting NP diagnostic accuracy.

Moreover, different cut-offs are indicated in chronic and acute settings. In outpatients with suspected HF, a plasma value less than 35 pg/mL for BNP and 125 pg/mL for NT-proBNP can exclude the diagnosis whereas in the emergency department, the cut-offs are 100 pg/mL and 300 pg/mL, respectively [27].

From a practical point of view, different aspects should be considered. In the chronic setting, the possibility of screening patients with suspected HF could decrease the need for more expensive diagnostic tools such as echocardiography and could help primary physicians better select patients in whom the diagnostic pathway for HF should be continued. However, NP measurements are not widely available. Moreover, given the greater prevalence of HF in older than younger patients and the frequent presence of comorbidities influencing NP serum levels [30], the number of patients in whom an HF diagnosis is ruled out by NPs might potentially be low. In the acute setting, in emergency departments, BNP and NT-proBNP measurements should ideally be easy to perform and should provide rapid results to avoid any delays in the appropriate therapy. Therefore, point-of-care testing might be used to optimize NP use [31,32].

## 5. Natriuretic Peptides for Stratifying Chronic Heart Failure Prognosis

Several studies have clearly demonstrated that plasma BNP and NT-proBNP levels are strong predictors of outcomes in both acute and chronic HF [3,33,34,35,36]. Plasma BNP provides prognostic information in patients with chronic HF (CHF) and patients with asymptomatic or minimally symptomatic LV dysfunctions [34]. Every 100 pg/mL increase in plasma BNP is associated with a 35% increase of a relative risk of death. In multivariable models, plasma BNP has been found to be a strong indicator of risk and may be a better predictor of survival than traditional risk factors such as the NYHA class and possibly the left ventricular ejection fraction (LVEF) [34]. Higher plasma levels of BNP are also associated with a poorer prognosis among patients with more advanced stages of CHF [36]. Moreover, the prognostic relevance of NT-proBNP and BNP can be further improved through their integration with other biomarkers: a multiparametric strategy has been demonstrated to improve prognostic stratification [37,38].

Notably, not only baseline levels, but also changes in BNP and NT-proBNP in response to HF therapy may be valuable in prognostication. A meta-regression analysis [39], including randomized trials with patients with CHF with decreased LVEF (HFrEF), evaluated the associations between BNP and NT-proBNP changes and the risk of HHF in patients with chronic HF. An increase in BNP and NT-proBNP was significantly associated with a greater risk of HF progression. In the Val-HeFT trials, the role of NT-proBNP changes related to prognosis was investigated by a post hoc analysis [40]. The results of the study clearly showed that a single determination of NT-proBNP had a greater prognostic value than both absolute and relative changes (the area under the curve from the ROC curve analysis was 0.702, 0.592 and 0.602, respectively). Interestingly, in the same study, a decrease during follow-up in NT-proBNP below the cut-off of 1000 pg/mL was found to be associated with a better prognosis as well as the persistence of a level below 1000 whereas an increased or a persistent NT-proBNP level above this cut-off was associated with a poorer prognosis [40,41]. According to this evidence, NPs may be useful both to stratify the prognosis and to understand the variations in the expected outcomes in response to HF therapy. Therefore, NPs have also been investigated to assess their utility in guiding therapy.

## 6. Use of Natriuretic Peptides to Guide CHF Therapy

As shown in Table 1 [42,43,44,45,46,47,48,49,50], several studies have sought to evaluate the use of NPs in guiding HF medical therapy. Troughton et al. [43] first demonstrated that treatment guided by plasma N-BNP concentrations, compared with a clinically guided treatment, decreased the total number of cardiovascular events. The baseline BNP concentrations were higher among patients who experienced clinical events than those who did not. A total of 75% of clinical events occurred among patients with a baseline N-BNP > 200 pmol/L. Analogously, in the PROTECT study on HF [48], NT-proBNP-guided care was found to decrease CV events. However, in the GUIDE-IT study [49,50], the only multicenter, controlled and randomized study reported to date, the strategy of NT-proBNP-guided therapy was not found to be more effective than a usual care strategy. The cardiovascular mortality rate was 12% in the biomarker-guided group and 13% in the usual care group (HR: 0.94; 95% CI: 0.65–1.37; *p* = 0.75). Notably, by 12 months, the median NT-proBNP concentration similarly decreased in the biomarker-guided and the usual care groups (from a median of 2568 pg/mL to 1209 pg/mL with a 53% decrease and from a median of 2678 pg/mL to 1397 pg/mL with a 48% decrease, respectively). The decrease in NT-proBNP achieved in both arms of the GUIDE-IT study exceeded those reported in most other studies of this type. Interestingly, in both groups, a low—but similar—percentage of patients received the dosages recommended by the treatment guidelines (≥ 50% of the target dose of β-blockers or angiotensin-converting enzyme inhibitors/angiotensin receptor blockers or any dose of mineralocorticoid antagonists). Given the similar treatments and similar decreases in NT-proBNP levels, the results of the study were neutral, as might have been expected. 

However, given that an optimal recommended treatment should be achieved independently of NT-proBNP levels, in daily clinical practice, BNP and NT-proBNP should be considered to be tools to monitor the efficacy of CHF therapy rather than to guide the therapy.

## 7. BNP and NT-proBNP in Acute Decompensated Heart Failure: Inpatient and Outpatient Utility

### 7.1. NPs during Admission for Acute Decompensated Heart Failure (ADHF)

The prognostic relevance of NT-proBNP has been demonstrated in patients admitted for acute decompensated HF [51]. Analogous to the findings for CHF, a decrease in this NP after admission generally reflects a favorable response to therapy and serves as a marker of better outcomes [52].

The relationship between plasma BNP levels at admission and the risk of in-hospital mortality in patients with acute decompensated HF was assessed in the ADHERE registry [51]. An elevated admission BNP level was a significant predictor of in-hospital mortality in acute decompensated HF with either a decreased or preserved systolic function, independent of other clinical and laboratory variables.

The relationship between BNP and NT-proBNP changes in response to HF therapy has also been investigated in ADHF in several studies [3]. In patients admitted for ADHF, the BNP and NT-proBNP levels were found to decrease with a hemodynamic improvement [53] because of the relationship between cardiac filling pressure and NP excretion. Therefore, BNP, owing to its shorter half-life than that of NT-proBNP, has been investigated as a parameter to evaluate patient response to diuretic therapy with standard clinical assessments (i.e., physical examinations as well as signs, symptoms and cardiac testing) [3].

Persistently high BNP serum levels may reflect the persistence of relevant clinical or subclinical congestion. The persistent elevation of plasma BNP before hospital discharge is predictive of death or readmission [52,53,54,55,56]. A systematic review—including one randomized trial, three quasi-experimental studies and forty observational studies—found low-quality evidence supporting an association between the achievement of natriuretic predischarge thresholds (e.g., BNP ≤ 250 pg/mL or NT-proBNP decline in at least 30%) and decreased acute HF mortality and readmission [55,56]. However, the BOT-AcuteHF [57] randomized controlled trial studying the effect of treating patients to achieve a target NP level found no improvement in outcomes with an NP-guided strategy. Nonetheless, patients who achieved the NP target had better outcomes than those who did not. One possible explanation for the failure of an NP-guided strategy to improve outcomes is that natriuretic targets are not achievable in several patients, given the limitations of current therapies [57]. BNP was compared between post-discharge visits and BNP at discharge.

The accuracy of BNP measurements in detecting residual congestion can be integrated with other tools reflecting congestion, thus providing a more reliable picture of the hemodynamic stability achieved before patient discharge. This finding is relevant, given the high risk of short-term readmission potentially associated with the inadequate treatment of congestion [58,59]. Among the examinations capable of integrating the information from NPs, ultrasound parameters or a bioimpedance vector analysis (BIVA) [54,60,61] may be particularly useful in daily clinical practice.

The use of point-of-care ultrasound technology beyond assessing cardiac structure and function may aid in evaluating central and peripheral congestion. Central venous congestion can be easily assessed by the detection of “thoracic comets”, i.e., thoracic B lines, whereas peripheral congestion can be assessed by estimating central venous pressure [62]. The persistence of B lines before discharge is associated with a greater incidence of HF readmission and mortality; therefore, routine evaluation upon admission and discharge may be used to guide decongestive strategies and aid in prognostication [63]. For a broader assessment of venous congestion, the inferior vena cava diameter as well as intrahepatic and intrarenal Doppler flow parameters have also been associated with organ congestion and impaired natriuretic responses to diuretic therapy [64].

A BIVA can serve as an alternative method for quantifying volume overload in acute settings and may provide objective measures to improve clinical decision-making and predict outcomes [59,60]. In a BIVA, bipolar electrodes are placed at the wrist and ankle and data are graphically displayed, thus enabling the short-term mortality risk and volume status to be accurately quantified [60]. A BIVA can provide indices of general cellular health, which have major prognostic implications, as well as total body volume. Knowledge of these parameters can provide insights into a short-term prognosis as well as the presenting volume status. A BIVA together with BNP can provide prognostic information for patients with HF as well as explain the 40% risk of death in these patients, independently of an acute or a chronic HF condition [54].

### 7.2. NPs and Patient Monitoring after Discharge

In the follow-up of patients after discharge, serial NP measurements have been investigated as a strategy for the early detection of hemodynamic progression and to avoid patient readmission.

Notably, the use of NPs for the early detection of HF progression should not be based on the recommended cut-offs for the diagnosis of new onset HF in acute and chronic settings [65]. As described above, these values are very low to maximize their sensitivity and negative predictive value, but are inadequate for detecting progression in patients with an established diagnosis of HF, whose serum levels typically exceed these cut-offs. The effective use of NPs to monitor hemodynamic stability in patients should instead be based on a reference value assessed in the presence of hemodynamic stability and which generally differs among patients [65]. For example, at discharge after ADHF with an optimal hemodynamic status, NT-proBNP levels might be approximately 500 pg/mL in one patient whereas another patient with a more advanced disease might have levels of approximately 1500 pg/mL. For both patients, the NT-proBNP values indicate the presence of a “dry” status. During follow-up, the reassessment of NT-proBNP and BNP could indicate HF progression if the measured values exceed those at the baseline, thus indicating a “wet” status. In this setting, changes in NP levels should be considered relevant if they exceed the natural variations in these biomarkers. An increase in levels accompanied by a > 30% relative change from “dry” BNP/NT-proBNP is generally considered to reflect changes in hemodynamic stability.

The HABIT trial [66] was a study designed to monitor the daily concentrations of BNP and to determine their correlations with ADHF. The study enrolled participants who either: (1) were admitted to the hospital with decompensated HF and had a BNP concentration of 400 pg/mL or an N-terminal B-type NP (NT-proBNP) concentration of 1600 pg/mL during admission; or (2) were seen in an outpatient setting with signs or symptoms of progression. Although NPs have known applications for monitoring and are prone after discharge, the effectiveness of daily monitoring has not been demonstrated and is subject to many variables often not corroborated by a need for a new therapist intervention. In addition, for the early detection of HF progression, NP accuracy could be improved by integrating echocardiography information [67]. The integrated management of BNP levels and the detection of echo-Doppler signs of elevated left ventricular filling pressure are associated with better outcomes and a lower incidence of renal function progression.

## 8. Natriuretic Peptides and New Modifier Drugs

### 8.1. Recent Randomized Trials and NPs

In recent years, trials evaluating the efficacy of new treatments in patients with HF have used the levels of NT-proBNP and/or BNP as inclusion criteria, as shown in Table 2. The use of NPs to enroll patients in trials has been mainly aimed at selecting patients at a higher risk of events to decrease the study sample size. As shown in a “real world” sample, the use of NP criteria enabled the identification of a group of patients with CHF at an elevated risk [68]. In contrast, the use of these criteria might limit the proportion of patients treated with new drugs [68,69]. However, major regulatory authorities (the Food and Drug Administration (FDA), the European Medicines Agency (EMA) and the National Institute for Health and Care Excellence (NICE)) have not suggested that the initiation of sacubitril/valsartan should be restricted to patients with elevated NT-proBNP. This suggestion is supported by the results of the subgroup analyses.

Baseline NT-proBNP levels were not significantly associated with drug treatment efficacy in trials demonstrating the efficacy of drugs in patients with LVEF < 40% or with LVEF > 40%. However, in the VICTORIA trial in patients with very high NT-proBNP, a lower benefit from vericiguat was observed whereas in GALACTIC-HF, omecamtiv mecarbil showed a trend toward more favorable effects in patients with higher rather than lower levels of NT-proBNP.

### 8.2. Sacubitril/Valsartan and NT-proBNP/BNP Changes

The advent of new therapies such as sacubitril/valsartan has substantially influenced the clinical course and prognosis of patients with HF as well as the use of NPs to monitor these patients. Sacubitril inhibits an endothelial endopeptidase, neprilysin, which is involved in the degradation of NPs [70] and may be overactivated in HF [5,71,72] (Figure 1). This inhibition increases NP activity [73], thereby causing two early opposite effects on the serum levels of BNP and NT-proBNP (Figure 2). Sacubitril/valsartan in comparison with the enalapril group is associated with an early increase in plasma BNP levels followed by a significant decrease [74]. In contrast, because of the favorable hemodynamic effects and subsequent reverse remodeling, NT-proBNP decreases early and has been found to diminish further during longer follow-up periods [75,76,77]. In summary, the levels of BNP reflect the action of the drug whereas the levels of NT-proBNP reflect the effects of the drug on the heart. Further confirmation was provided by the finding that 1 month after randomization, 24% of patients with an NT-proBNP level > 1000 pg/mL had a decrease to < 1000 pg/mL after a valsartan/sacubitril treatment compared with enalapril. Furthermore, NT-proBNP levels fell below 1000 pg/mL in 31% of patients treated with valsartan/sacubitril. In contrast, only 17% of enalapril-treated patients achieved a similar result.

The decrease in NT-proBNP levels in patients treated with valsartan/sacubitril has been correlated with a better cardiac performance, as demonstrated by improvements in several echocardiographic parameters [78].

### 8.3. SGLT2i and NP Changes

SGLT2i has been demonstrated to positively influence the natural history of patients with HF with LVEF either below or above 40% [9,10,11,12]. The mechanisms through which SGLT2i decreases the risk of HF-related events have not been fully elucidated although the effects on patient prognosis are striking [78,79,80,81,82]. Interestingly, the DEFINE-HF study [83] assessed 263 patients randomized to a treatment with 10 mg dapagliflozin daily vs. a placebo for 12 weeks and evaluated the changes in NT-proBNP. In the DAPA-HF study [84], treatment with dapagliflozin, compared with a placebo, was found to result in a statistically significant mild decrease in NT-proBNP at 8 months. These results may be associated with the slower and weaker effect of SGLT2i on NT-proBNP serum levels; the favorable effect of SGLT2i may be due to its action on different pathophysiologic pathways from those targeted by sacubitril/valsartan.

In the EMPEROR study, patients with HFrEF were randomly assigned to a placebo or 10 mg empagliflozin daily. NT-proBNP was measured at the baseline and at 4 weeks, 12 weeks, 52 weeks and 100 weeks. Empagliflozin decreased the risk of major cardiorenal events without heterogeneity across the NT-proBNP quartiles. Moreover, empagliflozin treatment significantly decreased NT-proBNP at all timepoints examined; by 52 weeks, the adjusted mean difference with respect to the values in the placebo group was 13% (*p* < 0.001). NT-proBNP in the lowest quartile (<1115 pg/mL) 12 weeks after randomization was associated with a lower risk of subsequent cardiovascular death or HF hospitalization regardless of the baseline concentration [85].

According to these results, SGLT2i significantly decreases NT-proBNP, but to a lesser extent than sacubitril/valsartan. However, baseline NT-proBNP serum levels are not associated with their beneficial effects. 

## 9. Current Guideline Recommendations Regarding the Use of BNP and NT-proBNP

Several differences exist among the current recommendations regarding the use of NT-proBNP and BNP in patients with HF. In the most recent ESC guidelines for HF, BNP and NT-proBNP are indicated only to rule out the diagnosis of suspected HF [27].

The ACC/AHA/HFSA guidelines [86] suggest that NT-proBNP biomarker screening and an early intervention may prevent HF (recommendation IIa-B-R). They also argue that in patients with dyspnea, an NT-proBNP assay is useful to support or exclude the diagnosis of heart failure (recommendation IA) and to establish the prognosis and severity of the disease (IA), especially in patients entering the hospital for exacerbations of heart failure. Furthermore, these guidelines indicate that NT-proBNP can be useful before discharge to identify a higher risk (recommendation IIA, B-NR). The usefulness of this biomarker as a guide for therapy is not well-defined.

The latest NICE guidelines [87] also recommend dosing NT-proBNP in people with suspected heart failure, but they also recommend undergoing a specialist evaluation with the execution of a transthoracic echocardiogram within 2 weeks for patients with NT-proBNP values > 2000 ng/L and within 6 weeks for patients with values between 400 and 2000 ng/L. These guidelines also point out that values below 400 ng/L make the diagnosis of heart failure less likely and, therefore, it is useful to look for other causes.

## 10. Conclusions

BNP and NT-proBNP play a key role in the current management of patients affected by HF by contributing to the identification of patients at a high risk of HF as well as to the diagnosis and prognostic stratification of patients already affected by it. Even if there is no evidence regarding the possible role of NPs in guiding HF therapy, they can be useful to evaluate the response to therapy in ADHF and CHF. Finally, over the last years, NT-proBNP and BNP have been included as inclusion criteria in randomized trials although their levels were not associated with a different efficacy of the treatments.

## Figures and Tables

**Figure 1 jcm-11-03192-f001:**
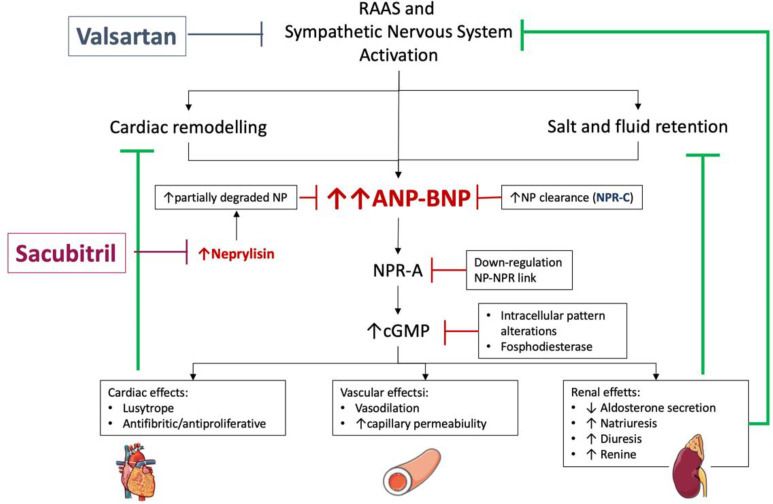
Effects of sacubitril/valsartan on the natriuretic peptide system. The inhibition of neprilysin by sacubitril increases the availability of active natriuretic peptides, which exert favorable effects by interacting with natriuretic peptide receptors. Arrows indicates the increase (↑) or the decrease (↓) of the different mentioned effects. ANP: atrial natriuretic peptide; BNP: brain natriuretic peptide; NP: natriuretic peptide; NPR-A: type A natriuretic peptide receptor; NPR-C: type C natriuretic peptide receptor.

**Figure 2 jcm-11-03192-f002:**
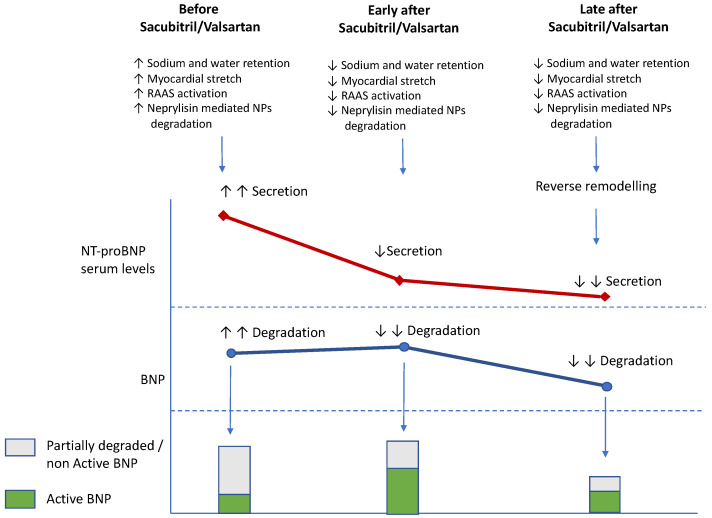
After sacubitril/valsartan administration, the inhibition of neprilysin leads to decreased degradation of BNP and to hemodynamic improvements. Consequently, it can be hypothesized that BNP serum levels increase and show a greater proportion of active hormones whereas NT-proBNP levels decrease. Late after the start of sacubitril/valsartan therapy, the hemodynamic improvement and the anti-remodeling effect lead to decreases in both BNP and NT-proBNP. Arrows indicates the increase (↑) or the decrease (↓) of the different mentioned effects. BNP: brain natriuretic peptide; NP: natriuretic peptide; NT-proBNP: amino-terminal brain natriuretic peptide; RAAS: renin angiotensin aldosterone system.

**Table 1 jcm-11-03192-t001:** The main studies evaluating the usefulness of natriuretic peptides to guide heart failure therapy.

Study	Patients (Age)	Therapeutic Target	NP Target	At Target (%)	Lower NP than Control	Results
**BNP-guided**						
**STARS-BNP** [42]	220 (65)	<100 pg/mL	Low	33	Yes	Reduction in HF death and hospital stay
**Troughton** [43]	69 (70)	<200 pmol/L	Low	N.a.		Reduction in total cardiovascular events
**NT-proBNP-guided**						
**BATTLESCARED** [44]	364 (76)	NT-proBNP increase 150 HF score > 2.0	No	Minority	Yes	Improved one-year survival; improved 3-year survival in < 75 yrs
**TIME-CHF** [45]	499 (77)	NT-proBNP <400 (<75 yrs) <800 (>75 yrs) NYHA class II	No	49	Yes	No effect on HF hospitalization, only a positive trend
**NorthStar****Adherence** [46]	921	NT-proBNP 1000 for shift from primary care to HF clinic	-	-	-	No improvement in adherence
**PRISMA** [47]	345 (72)	Increase after discharge > 10% NT-proBNP > 850	High	80	No	No effects on days of admission and HF hospitalization
**PROTECT** [48]	151 (63)	NT-proBNP < 1000	Low	44	No	Decrease in events
**GUIDE-IT** [49,50]	1100 (75)	NT-proBNP < 1000	No	N.a.	No	Decrease in events

HF: heart failure; BNP: brain natriuretic peptide; NP: natriuretic peptide; NT-proBNP: amino-terminal brain natriuretic peptide.

**Table 2 jcm-11-03192-t002:** Inclusion criteria of the more recent randomized trials evaluating treatment of HFrEF. The criteria based on natriuretic peptides are reported.

Trial (Treatment)	Main Inclusion Criteria	Main Exclusion Criteria	NT-proBNP/BNP Inclusion Criteria				Enrolled/Screened Patients (Excluded for NPs)
**PARADIGM-HF (Sacubitril/Valsartan vs. Enalapril)**	HFrEF NYHA II–IV LVEF ≤ 35% (≤ 35%) Optimal treatment	SAP < 100 eGFR < 30 K > 5.2		hHF (last 12 months)			8442/10,513 (n.a.)
				Yes	No		
			BNP	≥100	≥150		
			NT-proBNP	≥400	≥600		
**PARAGON-HF** (Sacubitril/Valsartan vs. Valsartan)	NYHA II–IV LVEF ≥ 45% Structural heart disease	SAP < 110 eGFR < 30 K > 5.2	NT-proBNP	hHF (last 12 months)			4822/10,359 (n.a.)
				Yes	No		
			SR	200	300		
			AF	600	900		
**DAPA-HF** (Dapagliflozin vs. Placebo)	HFrEF NYHA II–IV LVEF ≤ 40% Optimal treatment	SAP < 95 eGFR < 30	NT-proBNP hHF (last 12 months)		If AF		4744/8134 (n.a.)
			Yes	No			
			≥400	≥600	≥900		
**EMPEROR-reduced**	HFrEF NYHA II–IV LVEF ≤ 40% Optimal treatment	SAP < 100 eGFR ≤ 20	NT-proBNP				3730/7220 (2603, 75%)
			LVEF	hHF (12 months)			
				Rhythm	Yes	No	
			36–40%	SR AF	≥600 ≥1200	≥2500 ≥5000	
			31–35%	SR AF	≥ 600 ≥1200	≥1000 ≥2000	
			≤30%	SR AF	≥600 ≥1200	≥600 ≥1200	
**EMPEROR-preserved**	NYHA II–IV LVEF > 40%	SAP < 100 eGFR ≤ 20		SR	AF		5988/11,583 (4353, 78%)
			NT-proBNP	>300	>900		
**SOLOIST** **(Sotaglifozin vs. Placebo)**	ADHF No O_2_, e.v. inotropes or vasodilators or diuretic	SAP < 100 eGFR ≤ 30	BNP	SR AF	≥150 ≥450		1222/1549 (85, 26%)
			NT-proBNP	SR AF	≥600 ≥1800		
**VICTORIA-HF** Vericiguat vs. Placebo	Recent hHF NYHA II–IV LVEF ≤ 45% Optimal treatment	SAP < 100 eGFR < 15	BNP	SR AF	≥300 ≥500		5050/6857 (1078, 70%)
			NT-proBNP	SR AF	≥1000 ≥1600		
**GALACTIC-HF** Omecamtiv Mecarbil vs. Placebo	NYHA II–IV LVEF ≤ 35% Optimal treatment	Mechanical support I.v. medication SAP < 85 eGFR < 20	BNP	SR AF	≥125 ≥375		8256/11,421 (1467, 46%)
			NT-proBNP	SR AF	≥400 ≥1200		
**AFFIRM-HF** Ferric Carboxymaltose vs. Placebo	ADHF LVEF < 50% Iron deficiency		BNP	SR AF	≥400 ≥600		1132/1525 (n.a.)
			NT-proBNP	SR AF	≥1200 >2400		

BNP: brain natriuretic peptide; NP: natriuretic peptide; NT-proBNP: amino-terminal brain natriuretic peptide; SR: sinus rhythm; AF: atrial fibrillation; LVEF: left ventricular ejection fraction; ADHF: acute decompensated heart failure; SAP: systolic arterial pressure; GFR: estimated glomerular filtration rate.

## Data Availability

Not applicable.

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
