# Peer review of "Brain Natriuretic Peptide Biomarkers in Current Clinical and Therapeutic Scenarios of Heart Failure"

_jcm, 2022, doi:10.3390/jcm11113192_

Round 1

Reviewer 1 Report

This is a very comprehensive review, it is broad based, yet relatively detailed. I congratulate the authors for approaching a topic that is of clinical interest to practicing physicians. They should be commended for their efforts on such a large project.

Only minor check required:

from a practical viewpoint - row 125

please don't use abbreviation without explaining it - CHF - row 136

row 191 - please insert here the abbreviation for acute decompensated heart failure 

PNs or NPs? - rows 308 and 388

Author Response

Reviewer 1

We would like to thank the reviewer for his/her comments. This is our point-to-point response to his/her comments

  • This is a very comprehensive review, it is broad based, yet relatively detailed. I congratulate the authors for approaching a topic that is of clinical interest to practicing physicians. They should be commended for their efforts on such a large project.

Response: thank you for the comments.

Only minor check required:

  • from a practical viewpoint - row 125

Thank you we’ve corrected the term

  • please don't use abbreviation without explaining it - CHF - row 136

Thank you, we’ve modified accordingly

  • row 191 - please insert here the abbreviation for acute decompensated heart failure 

Thank you, we added it.

  • PNs or NPs? - rows 308 and 388

Thank you, we changed PNs into NPs.

Reviewer 2 Report

To the editors and authors:

Alcidi et al. described a review article entitled brain natriuretic peptide biomarkers in current clinical and therapeutic scenarios of heart failure. This was well written and informative for the readers. However, there are few updated information regarding the topic compared to previous articles. Following are things I would like the authors to consider.

(1) What is the research question?

In the abstract, why don' t you revise the description adding that the authors would like to emphasize. Furthermore, in the introduction section, the authors need to add the reason why the authors decide to develop this article.

(2) Factors associated with elevated these markers.

As you may know, cardiac as well as non-cardiac factors are known to be  determines for these markers. It would be helpful if the authors describe explanations especially why non-(extra) cardiac factors may be associated with elevated these markers.

(3) Target value of these markers

In clinical practice, physicians would like to know target value of these markers. Sometimes there is an argument whether either absolute change (post minus pre) or relative change( pre divided by post) is better for monitoring the patients. It would be helpful, if the authors add the descriptions related to this issue.

Author Response

We would like to thank the reviewer for his/her comments. This is our point-to-point reply.

(1) What is the research question? In the abstract, why don' t you revise the description adding that the authors would like to emphasize. Furthermore, in the introduction section, the authors need to add the reason why the authors decide to develop this article.

Reply:

We thank the reviewer for the comment. The aim of our review was to offer a point of view for the routine use of natriuretic peptide serum levels in the routine clinical practice. In fact, although many years ago the natriuretic peptides have been available in the clinical practice, their use is still limited due to both an inadequate knowledge among physicians and the controversial results in some clinical settings.

This is the background of our paper which we stated in the abstract and introduction section by adding the following sentences:

Abstract, lines 19-20:

Nevertheless, the use of natriuretic peptides is still limited in clinical practice due to the controversial aspects of their use in the different clinical settings

Introduction, lines 43-44:

Nevertheless, in the routine clinical practice, the use of NPs is still limited by some controversial aspects. The purpose of this review is to focus on the main issues associated with the dosage of BNP and NT-proBNP serum levels in the management of patients with HF in order to improve their use in the routine clinical practice.

(2) Factors associated with elevated these markers. As you may know, cardiac as well as non-cardiac factors are known to be  determines for these markers. It would be helpful if the authors describe explanations especially why non-(extra) cardiac factors may be associated with elevated these markers.

Reply:

We tried to better focus on the factors influencing natriuretic peptides levels in the paragraph 2.2, lines 68-81:

NPs serum levels are also by age, the greater the age the higher their serum levels [18]. Serum levels are also influenced by an altered cleavage [18]. Particularly when BNP is considered, a greater body mass index is associated with its lower serum levels. This is due to the fact that the clearance of BNP is associated with the activity of endopeptidase and the type C NP receptor, which are present mainly in adipose tissue—particularly in the obese population [4]: For this reason, these patients may have lower plasma BNP values than the normal population. On the other hand, tthe clearance of BNP and NT-proBNP is also mediated by renal excretion. In patients with chronic kidney disease, decreased estimated GFR is associated with elevated plasma BNP and an even greater elevation in NT-proBNP concentrations, whose clearance is more dependent on renal excretion.

Finally, beyond the parameters associated with cardiac function and filling pressure [19], atrial fibrillation (AF) is associated with elevated NPs.

These aspects should be considered together to optimize the use of serum levels NPs in routine clinical practice.

(3) Target value of these markers. In clinical practice, physicians would like to know target value of these markers. Sometimes there is an argument whether either absolute change (post minus pre) or relative change( pre divided by post) is better for monitoring the patients. It would be helpful, if the authors add the descriptions related to this issue.

 Reply:

We thank the reviewer for his/her helpful comment. This is a very relevant issue. We’ve underlined the relevance of a NT-proBNP less than 1000, which is generally considered a marker of response to heart failure treatment.

However, also the relative reduction of NT-proBNP and BNP could play a relevant role to stratify prognosis after treatment.

We better discussed this point in the paragraph 5, lines 158-168:

An increase in BNP and NT-proBNP is significantly associated with greater risk of HF progression. In the Val-HeFT trials, the role of NT-proBNP changes related to prognosis has been investigated by a post-hoc analysis [40]. The results of the study clearly showed as a single determination of NT-proBNP showed a greater prognostic value than both absolute and relative changes (area under the curve at ROC curve analysis 0.702, 0.592 and 0.602, respectively). Interestingly, in the same study, a decrease during follow-up in NT-proBNP below the cut-off of 1000 pg/ml has been found to be associated with better prognosis as well as the persistence of a level below 1000, whereas an increase or a persistently NT-proBNP above this cut-off has been associated with poorer prognosis [40–41]. According to this evidence, NPs may be useful both to stratify prognosis and to understand variations in expected outcomes in response to HF therapy.

Round 2

Reviewer 2 Report

Dear Editor and Authors

I have no further comments to revise.